# Bioprinting of Cartilage with Bioink Based on High-Concentration Collagen and Chondrocytes

**DOI:** 10.3390/ijms222111351

**Published:** 2021-10-21

**Authors:** Evgeny E. Beketov, Elena V. Isaeva, Nina D. Yakovleva, Grigory A. Demyashkin, Nadezhda V. Arguchinskaya, Anastas A. Kisel, Tatiana S. Lagoda, Egor P. Malakhov, Valentin I. Kharlov, Egor O. Osidak, Sergey P. Domogatsky, Sergey A. Ivanov, Petr V. Shegay, Andrey D. Kaprin

**Affiliations:** 1A. Tsyb Medical Radiological Research Centre—Branch of the National Medical Research Radiological Centre of the Ministry of Health of the Russian Federation, 10 Zhukova, 249031 Obninsk, Russia; kusimona@yandex.ru (E.V.I.); yakovleva.40@mail.ru (N.D.Y.); dr.dga@mail.ru (G.A.D.); nnv1994@mail.ru (N.V.A.); ki7el@mai.ru (A.A.K.); la-goda@yandex.ru (T.S.L.); malaxowegor@yandex.ru (E.P.M.); oncourolog@gmail.com (S.A.I.); 2National Research Nuclear University Mephi Obninsk Institute for Nuclear Power Engineering, 1 Studgorodok, 249039 Obninsk, Russia; 3Federal State Autonomous Educational Institution of Higher Education I.M. Sechenov First Moscow State Medical University of the Ministry of Health of the Russian Federation, 8-2 Trubetskaya, 119991 Moscow, Russia; 4Kvantorium Technology Park, 131 Lenina, 249035 Obninsk, Russia; vkharlov.obninsk@gmail.com; 5Imtek Ltd., 15A 3rd Cherepkovskaya, 121552 Moscow, Russia; eosidak@gmail.com (E.O.O.); Spdomo@gmail.com (S.P.D.); 6Gamaleya Research Institute of Epidemiology and Microbiology Federal State Budgetary Institution, Ministry of Health of the Russian Federation, 18 Gamalei, 123098 Moscow, Russia; 7Russian Cardiology Research and Production Center Federal State Budgetary Institution, Ministry of Health of the Russian Federation, 15A 3rd Cherepkovskaya, 121552 Moscow, Russia; 8National Medical Research Radiological Centre of the Ministry of Health of the Russian Federation, 4 Koroleva, 249036 Obninsk, Russia; dr.shegai@mail.ru (P.V.S.); kaprin@mail.ru (A.D.K.)

**Keywords:** biofabrication, bioprinting, hydrogel, collagen, bioink, scaffold, chondroblasts, chondrocytes, cartilage

## Abstract

The study was aimed at the applicability of a bioink based on 4% collagen and chondrocytes for *de novo* cartilage formation. Extrusion-based bioprinting was used for the biofabrication. The printing parameters were tuned to obtain stable material flow. In vivo data proved the ability of the tested bioink to form a cartilage within five to six weeks after the subcutaneous scaffold implantation. Certain areas of cartilage formation were detected as early as in one week. The resulting cartilage tissue had a distinctive structure with groups of isogenic cells as well as a high content of glycosaminoglycans and type II collagen.

## 1. Introduction

Cartilage is a connective tissue. It is devoid of nerves, blood, lymphatic vessels, and has a low density of cells, which are mainly represented by chondrocytes [1,2]. Cartilage is a part of the synovial joints, spine, ribs, outer ears, nose, and airways. Due to the low inherent potential of cartilage, the tissue’s pathologies remain an urgent medical and social problem. The tissue can be damaged in the result of physical injury, congenital anomalies, natural aging, and surgical interventions (for example, in the case of cartilage integrity violation in the course of radical surgery among the patients with tumors). Medical procedures, grafting, and drug treatment are used to treat defects and diseases of the cartilage [3]. However, these measures are usually limited and not fully adequate. Tissue engineering is the most promising option in this regard.

There is a significant number of published data on restoring articular cartilage [4], the trachea [5], intervertebral disc [6], auricle [7], and thyroid cartilage [8]. Collagen is the most applied biomaterial for tissue-engineered structures (scaffolds) [9]. It is a structural protein that forms the basis of the body’s connective tissue and ensures its strength and elasticity. Collagen has a high biocompatibility, low immunogenicity, and regulates adhesion, migration, and differentiation of cells. It has a fibrous structure and is highly compatible with other materials [10,11,12,13]. It is known that collagen concentration affects the printing accuracy and shape retention of a scaffold after the printing, and more concentrated formulations of the material are preferable [14]. The application of collagen hydrogel in concentrations as high as 1.75 [14], 2 [15], and 2.4% [16] have been already described. Despite the promising results related to the scaffold’s short-term geometry preservation, the material itself remained quite fragile. Thus, traditional protocols involving the use of low or relatively low material concentrations do not meet the key requirement of bioprinting related to long-term maintenance of a scaffold’s specified geometric shape. The following strategies can be applied to improve the mechanical properties: (1) supplementing collagen with cross-linking agents (for example, riboflavin [17], or methacrylamide and methacrylate groups [18]); (2) creating multicomponent bioink [12,19,20]; or (3) using higher collagen concentrations. The applicability of collagen-based hydrogels with a concentration as high as 4% has been shown in a few studies [8,21].

The issue of nutrient and gas transport in and out of a scaffold remains one of the main tissue engineering problems [22]. Both in vitro and in vivo, the cells inside a scaffold actively consume oxygen, making the supply by diffusion alone insufficient. The lack of oxygen leads to a decrease rate of cell proliferation, their transition to rest state, and presumable death. The issue has been described consistently in a previous study [23]. A number of researchers have pointed out that it is possible to avoid cell death by adding to the scaffold cavities and channels (vessels). 3D-bioprinting is an appropriate tool for this purpose [24]. A simple solution to the problem is the use of scaffolds with a central channel. Such an approach was applied in the present study. The channel reduces the maximum depth of the diffusion path for the cells located in the scaffold farther away from its free surfaces. The depth (maximal diffusion distance) is equal to 2 mm for a cube-shaped scaffold with dimensions of 4 × 4 × 4 mm. For the scaffold with a 0.8 × 0.8 mm central channel, the value decreases to 800 µm.

This work aimed at the applicability of bioink based on 4% collagen and chondrocytes to form cartilage *de novo.* To achieve this goal, it was necessary to establish the optimal parameters of bioprinting with 4% collagen hydrogel, to obtain and characterize the primary chondrocyte culture, and to explore the potential of cartilage formation after implantation of the scaffold in an animal.

## 2. Results

### 2.1. Optimal Printing Parameters for 4% Collagen Hydrogel

Two parameters that supposed to be crucial for 4% collagen printability were layer height and output of the material with respect to calculated value. The results obtained showed that the use of the standard value of the material output (100%), regardless of the layer height, is poorly applicable and results in serious geometric imperfection of the printed objects. The same is fairly true for a minimum layer height (25% of the nozzle diameter). Material output increase up to 150% was not effective for the last case. The objects’ geometric shape invariably showed deterioration after just three to four printing layers (Figure 1). Thus, the printing parameters appropriate for 4% collagen hydrogel in the case of 21G needles should include a layer height of at least 50% of the nozzle diameter with the output of the material that is 50% greater than the calculated value.

### 2.2. Cell Source Features

The cells were derived from the costal cartilage of newborn rat pups, which provided high cell content (Figure 2A). After the primary isolation cell, viability was 95% and the total number of the cells was ~48 × 10^6^. The viability of zero passage cells used for the experiment was 97.9%. Since cell concentration in the bioink was ~16 × 10^6^ mL^−1^, each scaffold of 61.44 mm^3^ (considering central channel) was supposed to contain ~962 × 10^3^ living cells and ~20 × 10^3^ non-viable cells.

To confirm that the cells were identified correctly, their ability of GAG production and their accumulation in the extracellular matrix was tested via staining with alcian blue. Figure 2B shows the results of the examination; distinctive blue staining indicates that the cell actively produces extracellular matrix with high GAG content within one week of the incubation.

### 2.3. In Vivo Research

The next day after the printing, the scaffolds maintained their geometric shape and the central channel was clearly visible (Figure 3A). The cells were uniformly distributed within the scaffold (Figure 3B).

The scaffolds (with surrounding tissues) derived from the animals on the 5th day after the implantation were easily observable (Figure 4A). The cells were unevenly distributed. Since the effect had not been observed before the implantation cell migration could be considered to occur. The implant was surrounded by a connective tissue capsule containing cells of the inflammatory infiltrate (mainly neutrophils). Connective tissue cells were also visible between the collagen fibers. Groups of cartilage isogenic cells were formed in the areas of the implant adjacent to the connective tissue capsule, which (presumably, along with fibroblasts) gave a positive nuclear reaction to PCNA (Figure 4B). At such an early stage, the scaffold already had a high but unevenly distributed accumulation of GAGs (Figure 4C) and type II collagen (Figure 4D). It can be generally noted that in five days after the implantation the scaffold loaded with chondroblasts (actively proliferating cells) and chondrocytes (mature cells) mixture showed a high production of specific metabolites. It provided the gradually replacement of the scaffold’s material with the corresponding extracellular matrix.

The signs of an inflammatory response still could be observed in 12 days after the implantation (Figure 5A). Groups of isogenic cells continued to form in the scaffold surrounded by a connective tissue capsule. In this area, morphological signs of proliferative activity (of different mitotic cycle stages) were observed and the nucleus of the chondroblasts gave a positive reaction to PCNA. The activity of chondroblasts within the scaffold remained irregular. The number of cells visualized by hematoxylin and eosin stain was low in the areas where the formation of isogenic cell groups (typical for cartilage) was not observed. It occurred in the areas of the pores and in places of scaffold material degradation. In these zones, the conditions (microenvironment) for chondroblasts proliferation (or their local concentration) were insufficient to trigger cartilage formation. Occasionally, chondroblasts emerged beyond the scaffold boundaries in the areas adjacent to the muscle tissue where cartilaginous tissue was also observed (Figure 5B). In one scaffold fragment, the formation of bone plates was found by means of Masson’s trichrome (Figure 5C).

Cartilage formation continued and was more evident by the 17th day (Figure 6). GAG accumulation was high. Its amount of variance among the animals could be supposed since there was some GAG accumulation decrease at the 26th and 34th days that was accompanied with active cartilage formation. The process was obvious at the day 26 when the tissue had certain cartilage specific features. Type II collagen accumulation was also high at the 17th day and did not distinctly change thereafter. In contrast, PCNA-positive cells were definitely more pronounced on day 17 than later. Moreover, small amount of PCNA-positive cells could be considered as an early evidence of cartilage formation completion since the native cartilage is characterized by relatively small amount of cells that do not proliferate (except perichondrium zone).

By the 40th day, the tissue had been formed (Figure 6). It had a high degree of homogeneity and adhered closely to the muscle fibers. The structure of *de novo* formed cartilage had typical cartilage tissue features: groups of isogenic encapsulated cells (chondrocytes) located in the central part, and peripherally oriented cells with an irregular (elongated) shape. The scaffold’s collagen was completely degraded and replaced with the cartilage tissue matrix. The formed cartilage tissue had a uniform and distinctively high accumulation of GAGs. Type II collagen abundantly accumulated in the central areas of the tissue.

## 3. Discussion

Collagen’s advantages and limitations relating to 3D-bioprinting are well known. In fact, collagen is the main option for all tissue engineering [9]. However, it has limited mechanical properties. Sodium alginate and gelatin methacryloyl (GelMA) have traditionally been used as alternatives applicable in cartilage regeneration. However, most alginate-based bioinks are inert hydrogels with a small amount of cell-adhesive ligands, which leads to a limited functionality [12]. GelMA is based on two components: gelatin and methacrylamide/methacrylate functional groups. Gelatin is a mixture of products of partial hydrolysis of various collagens (predominantly types I and III) [13]. The material is species-specific and donor-specific. Thus, the use of this hydrogel requires a careful assessment of its cytocompatibility [25]. On the other hand, the toxicity of methacrylamide and methacrylate functional groups is not sufficiently studied, but the fact of GelMA’s toxicity (at least, to some degree) is beyond doubt (judging by the properties of the initial reagent (methacrylic anhydride, CAS 760-93-0)). Therefore, the use of type I collagen in high concentration can be considered as the most reasoned decision, which significantly improves both the hydrogel’s rheological properties during the extrusion and the mechanical properties of the resulting gels [21] while also maintaining high biocompatibility [2,21].

For the bioprinting, collagen has been generally used in low concentrations (up to 1%). High gel concentrations have been rarely used (primarily due to expected decrease in nutrient and oxygen diffusion and the subsequent cell death). This assumption is primarily applicable to long-term incubation under static in vitro conditions since the diffusion is the only way of nutrient transport [26]. The main issue is related to oxygen. It is well known that oxygen diffusion is insufficient in a depth more than 200 μm [27]. The limitation of nutrient diffusion results in uneven cell activity, predominantly in outer surfaces of a scaffold [28]. The only way to improve the transport of nutrients and gases is through the use dynamic cultivation systems (e.g., bioreactors) that mimic the vascular system [29]. The issue has been discussed earlier [23].

Cell viability, proliferative activity, and chondrogenic potential dramatically depend on the donor’s age [30]. We used highly viable (>95%) costal cartilage cells derived from newborn rat pups and cultivated them after just a few days. According to R. Okubo et al. [31], the viability chondrocytes is a key factor that determines the chondrogenic stimuli sensitivity and the potential for cartilage formation in vivo. Therefore, it is crucial to maintain high cell viability rather than promote chondrogenesis during the cultivation.

In some samples obtained from an animal just 12 days after the implantation, in addition to cartilage, the formation of bone tissue has been observed. The bone plate formation was irregular and was not observed on histological slices later (34–40 days). It was probably caused by the molecular mechanisms of chondrogenesis regulation in a certain animal. N. Adachi et al. [32] observed cartilage ossification in some patients after replacing knee cartilage defects with scaffolds based on 3% type I atelocollagen with autologous chondrocytes. The authors supposed that, among other factors, the size of the defect and the number of chondrocytes in the gel could be the causes of the revealed effect.

In many studies reporting successful cartilage formation in animals or humans, tissue scaffolds were incubated in vitro from several days to several weeks prior to the implantation to ensure their maturation and the best results were supposed to be up to a three-week period [31,32,33,34,35]. Chondrogenesis depended on the cell density in the scaffolds (the higher the better), the nutrient medium composition, and the combination of growth factors, as well as the scaffold material. Maturation of scaffolds prior to implantation does not necessarily improve chondrogenesis [31]. In the case of 4% atelocollagen, prolonged pre-cultivation under static conditions may not yield a positive result due to poor diffusion of oxygen and other metabolites what was discussed above.

Despite its rapid development, extrusion-based bioprinting continues to be an experimental technology [36,37]. Precision bioprinting involves many factors, and a stable material output (filament extrusion) is the key one. The presence of air bubbles in a hydrogel or a bioink is one of the most common obstacles. Actually, it is hardly possible to remove all of the air bubbles at the stage of bioink components mixing. This factor can be deemed inevitable, however, and has a weaker effect in the case of self-assembled collagen than, for instance, in photopolymerizable GelMA. In the latter, the presence of the air is a factor of gel polymerization efficiency [25]. Another distinctive feature of collagen-based hydrogels is relating to their polymerization (which extends over time). It starts immediately after mixing of bioink components (collagen and neutralizing buffer) and continues during the printing and the subsequent scaffold incubation. The process significantly increases when the temperature rises. As a result, printing of a hollow object reveals hydrogel’s inherent type of scaffold’s geometry violation: the 90° angles set by the G-code were often smoothed out (rounded). The effect was not typical for the first layers, but was pronounced in the middle of the printing process. After the printing completion, the geometry impairments were less obvious.

The study actually was not aimed to reveal whether the scaffold vertical channel improve cell viability or not. The channel improved the diffusion of nutrients by reducing the maximum depth at which the cells were located. According to the published data, it is known that few cells can survive at distances exceeding 200 μm from the blood vessel (or artificial vessels with a nutrient medium) and that cells from tissues with a high oxygen consumption rate, such as the heart, pancreas, and liver, die when the diffusion distance exceeds ~100 μm [38]. Cartilage cells are generally more resistant in this regard and can survive in over 1 mm thick grafts [22,38]. We supposed that 4% collagen had a worse ability to provide the cells located deep in the scaffold with nutrients and gases than the 1% collagen which is typically applied. Thus, there was a need to level the possible effect and the vertical channel was added to the scaffolds. At the same time, such a precaution may be considered as excessive to some extent in view of data on human dermal fibroblasts cultured for 28 days inside collagen scaffolds [39]. It was shown that collagen concentration doesn’t influence cell viability (more than 90% after 28-day incubation), its morphology, and expression of extracellular matrix components.

Either differentiated cells or mesenchymal stem cells (MSCs) are used in tissue engineering. The use of mature, differentiated cells does not require additional bioink components responsible for cell differentiation direction. At the same time, the prospects for the clinical use of differentiated cells are not obvious. In the case of cartilage, the problem is that in cells, low density in the tissue (of an adult human), degenerative age processes, and a small ratio of chondroblasts are capable of activating proliferation [40,41]. Long-term in vitro cultivation may result in the loss of the cell’s specific phenotype. It takes just two to three passages to transform the cells into fibroblasts [41]. Thus, it is difficult to create scaffolds containing the patient’s own chondroblasts (more precisely, with a mixture of chondroblasts and chondrocytes). The present study was conducted to examine whether it is possible to use high-concentration collagen as the bioink basis to form cartilage *de novo*. The primary culture of chondrocytes applied in the research can be considered as an “ideal” condition. In the further study, MSCs will be applied due to their incomparably greater potential in terms of the future clinical application of bioprinting technology.

## 4. Materials and Methods

### 4.1. Chondrocyte Culture

The primary culture of chondrocytes was obtained in accordance with A. Gartland et al. protocol [42]. The procedures described below were conducted under the permission of the Bioethical Commission on Keeping and Using Laboratory Animals of A. Tsyb MRRC No. 1-N-00006 dated 05/07/2021. The rib cages of four and eight-day-old euthanized rat pups (fourteen animals) were obtained under sterile conditions, and the costal cartilage was isolated and washed with PBS (hereinafter PanEco (Moscow, Russia), unless otherwise stated). To remove all soft tissues, the cartilage was placed in a DMEM medium with 4.5 g/L glucose, 0.25% trypsin and 0.2% type I collagenase (Gibco, Grand Island, NY, USA) and were stirred moderately for 30 min at +37 °C. The solutions were renewed, and the step was repeated. Afterwards, the samples were put for 30 min in 0.25% trypsin solution and then were centrifuged at 100× *g* for 2 min. The sediment (rib cartilage) was placed in 0.15% collagenase solution and was incubated overnight at +37 °C in a CO_2_ incubator. The next day, the solution with the sample was stirred for 30 min at minimum speed to separate the cells. Not-split tissue fragments were removed by filtration (100 nm, nylon, SPL Lifesciences, South Korea). The filtrate was centrifuged at 400× *g* for 5 min to inactivate the enzymes. The culture medium with 10% fetal bovine serum (Biosera, Nuaille, France) was added to the sediment and the step was repeated. The pellet was resuspended in the medium. Cell viability was analyzed by staining with 0.4% trypan blue solution in PBS. The cells were seeded to 75 cm^2^ flasks (Corning, NY, USA) with DMEM medium containing 10% serum, penicillin-streptomycin (50 U/mL and μg/mL, respectively), and glutamine (649 μg/mL). The cells were cultivated until a monolayer was formed (about four days), and then it was removed from the plastic and used for bioink preparation.

Some cells were cultured in 3.5 cm Petri dishes (Corning, NY, USA, 300 × 10^3^ cells per dish) to identify chondroblasts that can form glycosaminoglycans (GAGs). The protocol described by M. Gosset et al. was applied [41]. The cells were cultured under standard conditions for seven days. The culture medium was subsequently removed and the monolayer was washed twice with PBS and fixed with 4% glutaraldehyde (PanReac, Spain) at room temperature. After washing with 0.1 N HCl, the cells were stained with 1% alcian blue solution (8GX, Sigma-Aldrich, Saint Louis, MO, USA, in 0.1 N HCl) at room temperature. The monolayers were washed twice with 0.1 N HCl, dried, and moistened again with 0.1 N HCl. The dishes were examined using Biomed 3 microscope (Biomed, Moscow, Russia) with ToupCam 3.1 camera (ToupTek, Hangzhou, China).

### 4.2. Extrusion-Based 3D-Bioprinting

Bioprinting was performed on Invivo 4D2 (Rokit, Seoul, Korea) using 1.68 firmware. The input printing model was sliced using NewCreatorK 1.57.63. The hydrogel or bioink was in a Luer-Lock glass syringe with 8.3 mm inner diameter. The material output on the first layer was increased to take into account the error of bed levelling due to fixed step value (100 μm) of the procedure. The printing speed was 5 mm/s. The dispenser temperature was set at +4 °C, and the printing table was set at room temperature. The printing was performed on sterile Petri dishes (Corning, NY, USA).

To determine the optimal printing parameters for 4% collagen hydrogel a model object was applied. The effect of the layer height (25, 50, 75, and 100% of needle’s inner diameter) and the percentage of material output (100 and 150% of the estimated number) was examined by printing of 10 × 10 × 4.626 mm parallelepiped’s walls. The wall width was equal to two nozzle’s diameters (≥1.028 mm). The object was formed in 36, 18, 12, and 9 layers, depending on the layer height. The printing quality was assessed visually.

The bioink was prepared using sterile acidic solution of type I atelocollagen from porcine tendons at concentration 80 mg/mL (Viscoll, Imtek Ltd., Moscow, Russia) according to the previously described method [21]. On the day of the experiment, the cells were removed from culture dishes with EDTA-trypsin solution. After staining with trypan blue, cell viability was estimated. The cells were centrifuged (400× *g*, for 5 min), and resuspended in 0.25 mL of serum-free DMEM medium. The cell suspension was mixed with 0.25 mL of collagen buffer solution (Tris-HCl, 0.3 M, pH 8.0). The received solution was kept at +4 °C; for 10 min and then was mixed with 0.5 mL of collagen to obtain a bioink with 4% collagen concentration. The final cell concentration was ~16 × 10^6^ mL^−1^. Before the printing, the bioink was kept at +4 °C, and the printer’s chamber was sterilized using built-in 254 nm UV-lamps. The targeted object (scaffold) for the printing had to be cube-shaped, with outer dimensions of 4 × 4 × 4 mm and with 0.8 × 0.8 mm hollow central (vertical) channel. The STL-file was created using FreeCAD 0.17. The printing was performed with a 21G needle (inner diameter—514 μm). Layer height was 333 μm (65% of the nozzle diameter). Object infill percentage corresponded to a material output value. The dishes with the printed objects were filled with a warm (+37 °C) medium. The printed scaffolds were incubated at +37 °C; and 5% CO_2_ for one day. The medium’s composition was as follows: DMEM with 4.5 g/L glucose, 10% serum, penicillin-streptomycin (50 units/mL and μg/mL, respectively), 649 μg/mL glutamine.

### 4.3. Implantation of Scaffolds into Animals

After one day of the incubation, the scaffolds were implanted in two-month-old female Wistar rats (~200 g). The total number of the animals was 12 (i.e., 2 animals on each of 6 time-points). All operations with the animals were conducted under ether inhalation anesthesia. The scaffolds were implanted in the withers area. The operating field was cut off and treated with 70% ethanol solution. An incision made with scissors and a scalpel was used to form a pocket under the skin where the implant was placed. The edges of the pocket were pulled together, the wound was sutured (Monocryl Poliglecaprone 25, Ethicon, Raritan, NJ, USA). The implantation site was marked with a colored thread of suture material. The seam was treated with 3% hydrogen peroxide solution. The second suture was applied to the skin and re-treated with hydrogen peroxide. A medical adhesive was applied on top to ensure better fixation. The area around the operating field was anesthetized with 0.5% novocaine. The rats were isolated from each other in separate cages. The surgical site was examined daily. The animals were euthanized at intervals of ~one week, starting from the fifth day after the implantation. The final time point (40th day) corresponded to available data on the material’s biodegradation period after subcutaneous implantation [43]. The implant (scaffold) with surround tissues were histologically examined. All procedures at this stage were conducted according to the Bioethical Commission on Keeping and Using Laboratory Animals (No. 1-N-00006 dated 05/07/2021). The whole experimental timeline is shown in Figure 7.

### 4.4. Histological and Immunohistochemical Studies

The scaffolds with surrounding tissues were fixed for 24 h in acidic Bouin solution (including 1.3% trinitrophenol (Sigma-Aldrich, Darmstadt, Germany), 40% formalin (hereinafter BioVitrum (Moscow, Russia), unless otherwise specified). After washing in 70% ethanol, standard histological preparation of the samples was performed, and then they were placed in a paraffin medium (Histomix). Paraffined slices of 5 μm thick were obtained with a microtome (RM2235, Leica, Nuslohe, Germany) and placed on silanized glasses (S3003, Dako, Carpinteria, CA, USA). Dewaxed slices were stained with hematoxylin and eosin, alcian blue (8GX, Sigma-Aldrich, Saint Louis, MO, USA), for histological studies. The bone tissue was visualized using Masson’s trichrome. The slices were dehydrated in alcohol, cleared in (ortho-)xylene, and embedded in Canadian balsam (Merck, Darmstadt, Germany).

Polyclonal rabbit antibodies to the nuclear antigen of PCNA proliferating cells (PA5-27214, 1:200, Invitrogen, Taiwan) and monoclonal rabbit antibodies to type II collagen (SAB4500366, 1:50, Sigma-Aldrich, UK) were used for immunohistochemical studies. Secondary goat antibodies were conjugated with horseradish peroxidase (ab205718, 1:1000, Abcam, Cambridge, UK). Immunohistochemical solutions were prepared in PBS. Following the immunohistochemical study protocol, dewaxed slices immersed in citrate buffer (pH 6.0) were boiled (5 min) before primary antibodies to PCNA and type II collagen were applied. Endogenous peroxidase was blocked in 3% hydrogen peroxide solution. The blocking buffer was supplemented with 2% serum of secondary antibody donors, 1% bovine serum albumin, and 0.1% Triton X-100. The samples were incubated in the primary antibody solution overnight in a humid chamber at +4 °C. After washing in PBS, secondary goat anti-rabbit antibodies were applied to the slices for 1 h at room temperature. Substrate peroxidase was detected using diaminobenzidine (Liquid DAB+, K3468, Dako, Carpinteria, CA, USA). The slices were dehydrated in alcohol, cleared in (ortho-)xylene, and embedded in Canadian balsam (Merck, Darmstadt, Germany). Histological slices were examined by AXIO Imager A1 microscope (Carl Zeiss, Oberkochen, Germany) and Power Shot A640 camera (Canon, Tokyo, Japan).

## 5. Conclusion

The present study reveals the effectiveness of the bioink based on 4% collagen and primary chondrocytes applicable in extrusion-based bioprinting regarding to cartilage formation in vivo. The cartilage tissue obtained *de novo* had a typical structure with groups of isogenic cells and was rich with GAGs and type II collagen. Thus, 4% collagen based hydrogel can be considered as a biomaterial with still high biocompatibility but with better printability and mechanical properties compared to collagen with a conventional (low) concentration.

## Figures and Tables

**Figure 1 ijms-22-11351-f001:**
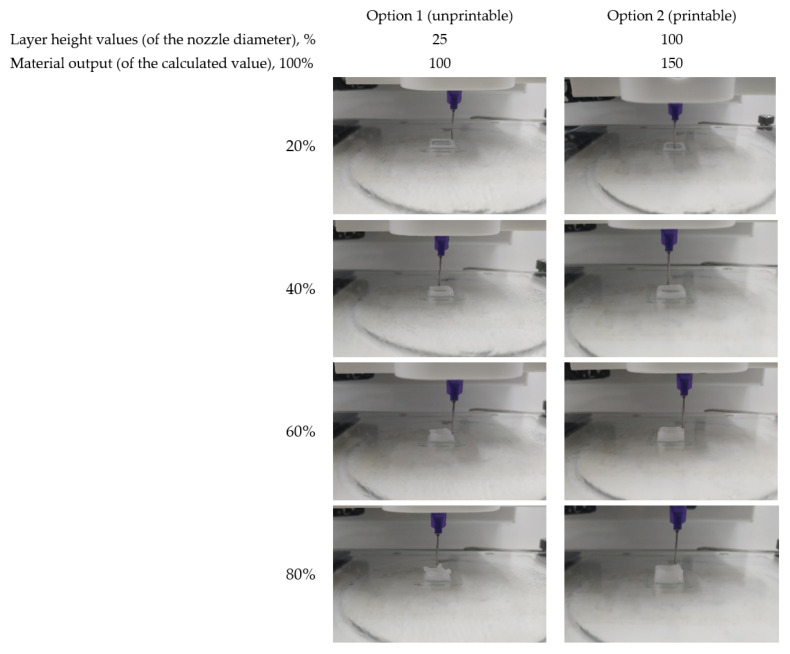
Determination of printable slicing settings. Printing process in the case of two options.

**Figure 2 ijms-22-11351-f002:**
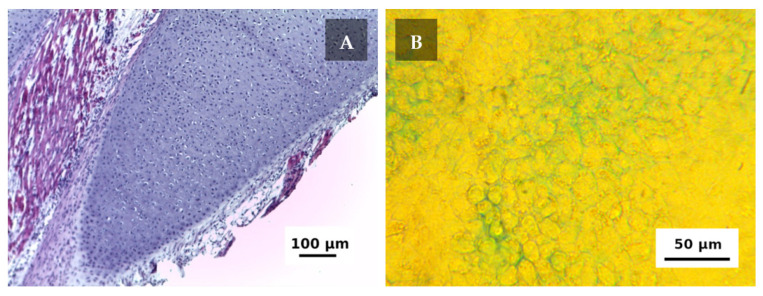
Cell’s source and the primary culture. (**A**) The native costal cartilage of a four-day-old rat pup with surrounding tissues stained with hematoxylin and eosin, objective lens ×10. (**B**) The cells of a zero passage after seven days of incubation stained with alcian blue, phase contrast, and objective lens ×25.

**Figure 3 ijms-22-11351-f003:**
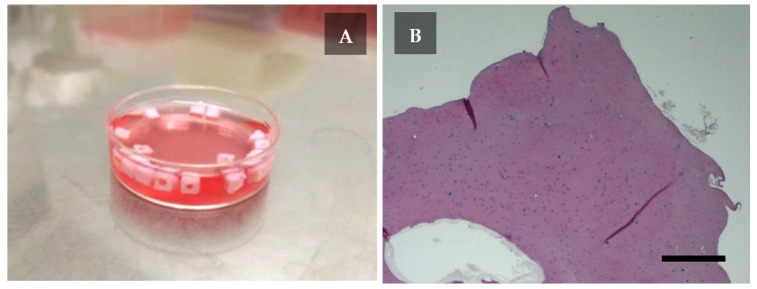
The cell-laden scaffolds after one day of incubation. (**A**) The scaffolds in the Petri dish. (**B**) Staining with hematoxylin and eosin. Objective lens ×2.5, scale bar—500 μm.

**Figure 4 ijms-22-11351-f004:**
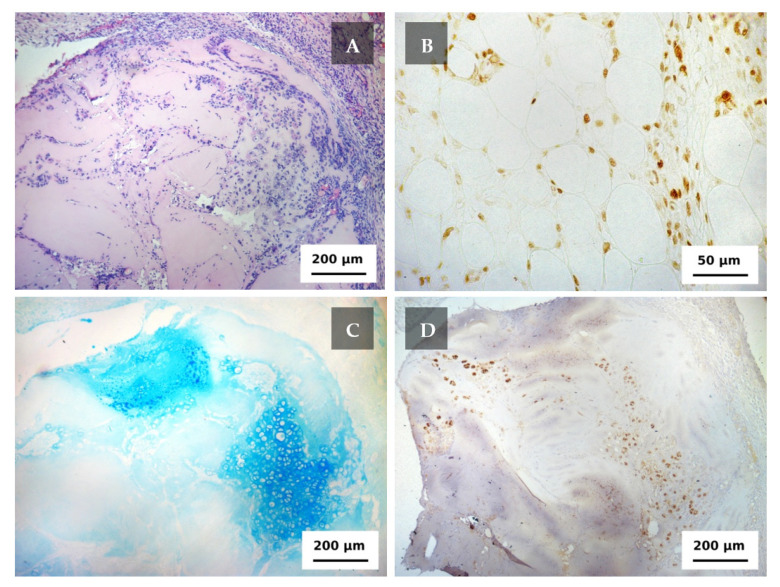
The scaffold in 5 days after the implantation. (**A**) Staining with hematoxylin and eosin. objective lens ×10. (**B**) PCNA-positive nuclei of chondroblasts and proliferating cells in the connective tissue capsule, objective lens ×40. (**C**) Staining for glycosaminoglycans, objective lens ×10. (**D**) staining for type II collagen, objective lens ×10.

**Figure 5 ijms-22-11351-f005:**
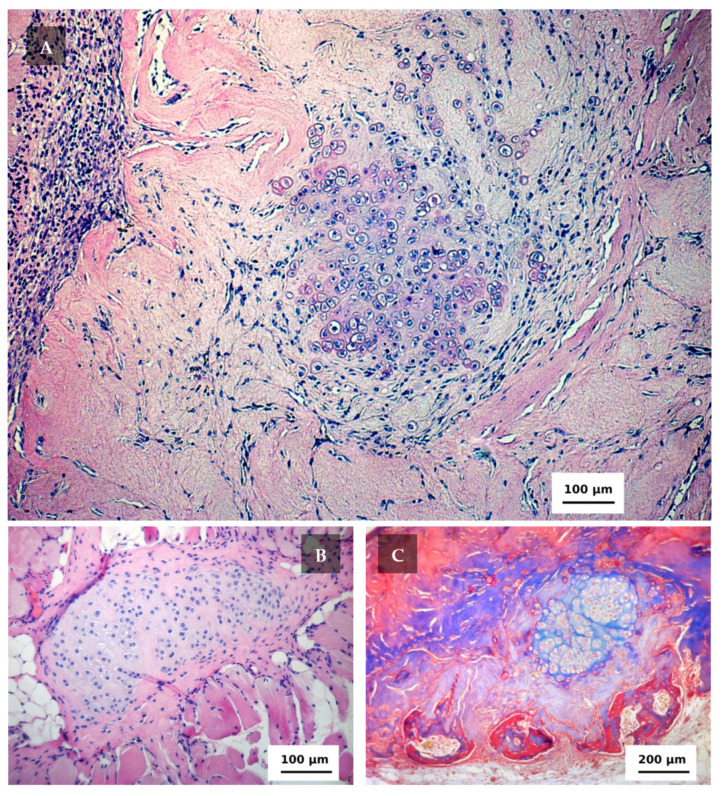
The cartilage tissue formed in 12 days after the implantation stained with hematoxylin and eosin. (**A**) Within the scaffold volume, objective lens ×10. (**B**) Among striated muscle tissue, objective lens ×20. (**C**) Bone tissue formation (in the lower part), Masson’s staining, objective lens ×10.

**Figure 6 ijms-22-11351-f006:**
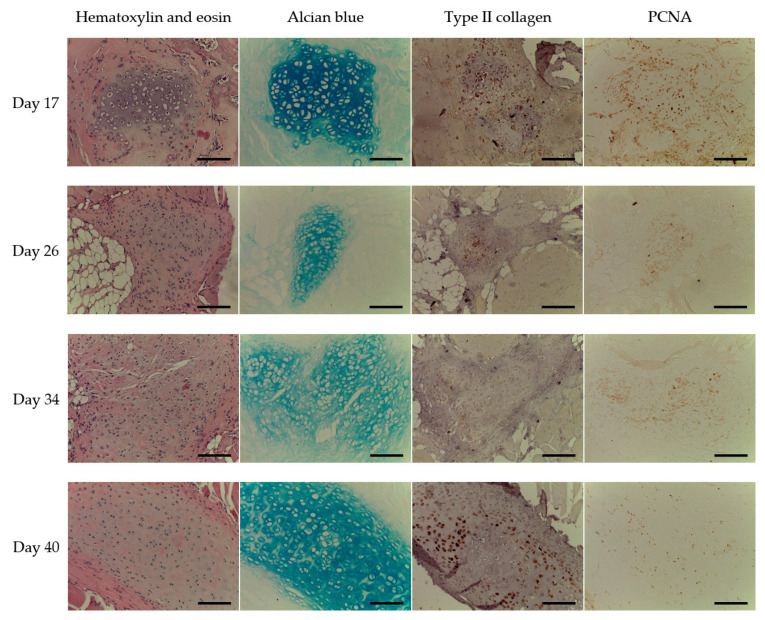
The cartilage tissue formed in 17–40 days after the implantation, objective lens ×20, scale bar—100 μm.

**Figure 7 ijms-22-11351-f007:**
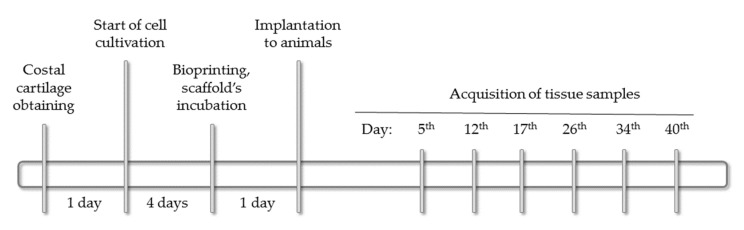
The study timeline.

## Data Availability

The study did not report any data.

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
