# Peer review of "Bioprinting of Cartilage with Bioink Based on High-Concentration Collagen and Chondrocytes"

_ijms, 2021, doi:10.3390/ijms222111351_

Round 1
Reviewer 1 Report
REVIEW REPORT
JOURNAL ID: International Journal of Molecular Sciences ISSN 1422-0067
MANUSCRIPT ID: ijms-1399786-peer-review-v1
TYPE: Article
Bioprinting of Cartilage with Bioink Based on High-Concentration Collagen and Chondrocytes
Beketov E.E. *, Isaeva E.V., Yakovleva N.D., Demyashkin A.G., Arguchinskaya N.V., Kisel A.A., Lagoda T.S., Malakhov E.P., Kharlov V.I., Osidak E.O., Domogatsky S.P., Ivanov S.A., Shegay P.V. and Kaprin A.D.
The manuscript addresses a modern and multidisciplinary research topic with an impact on therapy of defects and diseases of the cartilage. The authors showed the technical feasibility of 3D printing method with bioink based on collagen hydrogel in order to create scaffolds with central channel. The originality of the research is represented by the fact that the cartilage de novo was obtained using bioink with higher collagen concentrations, based on 4% collagen and chondrocytes. Also, the optimal parameters of extrusion-based 3D-bioprinting were tuned to obtain stable material flow.
Strengths of manuscript
The manuscript corresponds to the stated purpose and objectives of the journal.
The title accurately reflects the content of the paper.
The abstract is structured as a short synopsis of the paper.
The introduction presents in detail the current state of knowledge on the approached subject and highlights why this research is important. It is define the purpose of the work and its significance. Also, the introduction includes 24 relevant references, out of which 20 are published after 2015.
The manuscript contains a complex, but correctly designed and technically sound research method. The research steps are clearly presented in the “Materials and Methods” chapter. The instruments, research equipment, software programs and reagents are described in sufficient details to allow another researcher to reproduce the results. The research method was developed by documentation on research method of other studies, in this chapter being included 3 references.
The performed analyses are appropriate. The research results are presented at higher standards, including 7 figures. The way of reporting the results was elaborated taking into account the steps of the research method. The images are significant and suggestive for the research subject. The results are presented concisely and systematically. All figures are elaborated according to authors’ guideline.
The discussions present an interpretation of the results in perspective of previous studies and of the aim of the study. The comparison of the research results was made taking into account over 20 other references. The findings and their implications are discussed in the broadest context possible.
The conclusions are presented in generally manner and highlight the research implications in obtaining of biological material used for bone defects therapy. They are interesting for the readership of the journal.
The references are in accordance with the studied topic. The manuscript contains 29 references, most representing studies published after the year 2015.
Weakness of the manuscript
-The limitations of the work are not highlighted.
Author Response
Dear Reviewer,
We are grateful for such a comprehensive examination of our manuscript, as well as the possibility of improving it. The manuscript has been considerably revised. We hope the corrected version of the paper is more suitable for publication in your journal.
Yours faithfully,
Research Team
Response to the comments:
Please, discuss how the use of bioreactors could increase the viability of cells in culture (before implantation).
The text has been modified relating to the issue. More comprehensively the problem has been discussed in our previous research (Isaeva E.V. et al. Cell and Tissue Biology. 2021, 15(5), 493-502). The use of bioreactors is an excellent way for in vitro maturation step of biofabrication. But according to the aim of the study this step was minimized and supposed not to influence. Moreover, the discussion section contains the data that doubts the need in such a process in the case of cartilage.
A specific characterization of the obtained scaffolds is completely missing, please add this section to the manuscript.
The manuscript is supplemented with data on the scaffold immediately after the printing.
The viability of chondroblasts in scaffold, one day after the bioprinting therefore just before implantation should be included since these data are of extreme interest.
In the present study we used 4×4×4 mm cube-shaped scaffold while Live-Dead Assay is applicable to thin (slice like) scaffold. The analysis of the scaffold of another type (slice like) would not be relevant due to different conditions for the cells in the case of these two options (implanted one and the one studied with the assay).
However, our own data on collagen’s cytocompatibility (for the same chondrocytes) has already been published earlier (Arguchinskaya N.V. et al. Int J Bioprint. 2021, 7(2), 104-113). The material shows good results despite its high concentration (and thus, its increased viscosity during the printing and elasticity of the scaffolds).
Moreover, collagen cytocompatibility of different concentrations (cell viability after in vitro incubation) has been studied abundantly recently, especially for short-terms (up to 1 week).
Figures caption letters have to be inside the figures and more details should be provided.
The figures were modified.
A more schematic description of time points considered should be added in method section.
The schematic description is added to the manuscript.

Reviewer 2 Report
This paper of Beketov E.E et al. sounds interesting, but some aspects need to be improved in order to suitable for this journal.
Please, discuss how the use of bioreactors could increase the viability of cells in culture (before implantation).
A specific characterization of the obtained scaffolds is completely missing, please add this section to the manuscript.
The viability of chondroblasts in scaffold, one day after the bioprinting therefore just before implantation should be included since these data are of extreme interest.
Figures caption letters have to be inside the figures and more details should be provided.
A more schematic description of time points considered should be added in method section.
thanks
Author Response
Dear Reviewer,
We are grateful for such a comprehensive examination of our manuscript, as well as the possibility of improving it. The manuscript has been considerably revised. We hope the corrected version of the paper is more suitable for publication in your journal.
Yours faithfully,
Research Team

Reviewer 3 Report
- The fluidity of the text should be improved.
- Provide the experimental design, number of animals, site of implantation, time points of scaffold recovery.
- Cell isolation and expansion are well described. However, the expansion rate and efficiency of the method are not reported.
- As the Authors performed a trypan blue assay, they should report the number of live / dead cells before bio-printing (they state the percentage of live cells is over 95% but no specific result is provided). - Provide an immunohistochemical documentation of type I collagen expression to be compared with type II collagen positivity.
- Since high type I collagen concentrations are expected to decrease the diffusion of nutrients and oxygen with subsequent cell death, a live/death cell assay should be provided to corroborate the efficacy of the methodology.
- It should be provided a histological analysis of the scaffold before implantation, since is essential to establish what was the starting point.
- At each time point, provide complex images with histology, PCNA immunostaining, Alcian blue staining, collagen type I and II, of the same samples.
Author Response
Dear Reviewer,
We are grateful for such a comprehensive examination of our manuscript, as well as the possibility of improving it. The manuscript has been considerably revised. We hope the corrected version of the paper is more suitable for publication in your journal.
Yours faithfully,
Research Team
Response to the comments:
The fluidity of the text should be improved.
The text was revised and corrected.
Provide the experimental design, number of animals, site of implantation, time points of scaffold recovery.
- Cell isolation and expansion are well described. However, the expansion rate and efficiency of the method are not reported.
- As the Authors performed a trypan blue assay, they should report the number of live / dead cells before bio-printing (they state the percentage of live cells is over 95% but no specific result is provided).
Materials and Methods section was supplemented with data related to experimental design.
Expansion rate of the cells were not assessed since its expansion was not scheduled. The objective was to preserve cells phenotype and its cultivation was minimized (just to eliminate unviable cells).
Information related to cell viability by the moment of the printing was added to results section.
Provide an immunohistochemical documentation of type I collagen expression to be compared with type II collagen positivity.
The used biomaterial is a pure type I collagen. It is a well-studied material for the purpose. Its biodegradation rate has already been examined (the information is added to the text in materials and methods section). The protein is not cartilage-specific and its accumulation could not be considered as any significant evidence of cartilage formation.
Type II collagen (as well as GAGs) productions is the main evidence of cartilage formation. Histological and immunohistochemical studies can be considered as the most reliable research method that provides direct proof of the process (its dynamics and the fact of completion). Therefore, other research methods (including col1 or col2 expression) were not applied due to their indirect nature.
At the same time, we realize the possible advantages of such an analysis and we are going to use them in our future studies related to the examination of de novo formed cartilage tissue comparing to native cartilage.
Since high type I collagen concentrations are expected to decrease the diffusion of nutrients and oxygen with subsequent cell death, a live/death cell assay should be provided to corroborate the efficacy of the methodology.
In the present study we used 4×4×4 mm cube-shaped scaffold while Live-Dead Assay is applicable to thin (slice like) scaffold. The analysis of the scaffold of another type (slice like) would not be relevant due to different conditions for the cells in the case of these two options (implanted one and the one studied with the assay).
However, our own data on concentrated collagen’s cytocompatibility using the method (for the same chondrocytes) has already been published (Arguchinskaya N.V. et al. Int J Bioprint. 2021, 7(2), 104-113). The material shows good results despite its higher concentration (and thus, its increased viscosity during the printing and elasticity of the scaffolds).
It should be provided a histological analysis of the scaffold before implantation, since is essential to establish what was the starting point.
The manuscript has been supplemented with the data.
At each time point, provide complex images with histology, PCNA immunostaining, Alcian blue staining, collagen type I and II, of the same samples.
The way of data presentation was modified for the late time points (17-40 days). But we suppose that data on earlier time points is more accessible as is since there is a need to use different scale as well as another type of staining.

Reviewer 4 Report
The manuscript reports cartilage bioprinting as a strategy for de novo tissue regeneration. The content is interesting but there are many areas that are lacking in terms of experimental design.
- More characterization of the bioprinted construct in vitro is needed. Cell viability, proliferation, distribution within the construct after printing should be evaluated.
- Instead of just showing 1 photo in figure 1, printing consistency has to be checked.
- No control (collagen construct without cells) for the in vivo experiment. Are the presence of chondrocytes critical?
- There is quantitative data for the in vivo study. Images based on one animal are not representative.
- Figure 3 has to be revised. It's unclear what it is showing.
- The language in the manuscript has to be improved significantly
Author Response
Dear Reviewer,
We are grateful for such a comprehensive examination of our manuscript, as well as the possibility of improving it. The manuscript has been considerably revised. We hope the corrected version of the paper is more suitable for publication in your journal.
Yours faithfully,
Research Team
Response to the comments:
- More characterization of the bioprinted construct in vitro is needed. Cell viability, proliferation, distribution within the construct after printing should be evaluated.
Data on cell distribution after bioprinting is added. At the same time cell viability and proliferation within the scaffold in vitro were not the study’s aims since the aspects have been examined earlier (Isaeva E.V. et al. Cell and Tissue Biology. 2021, 15(5), 493-502).
- Instead of just showing 1 photo in figure 1, printing consistency has to be checked.
The whole conception of the figure has been revisited.
- No control (collagen construct without cells) for the in vivo experiment. Are the presence of chondrocytes critical?
Yes, type I collagen is a unique material with the excellent biocompatibility. At the same time this material does not provide any tissue specificity. The use of the material (whether cell-free or as MSC-laden option) do not provide formation of any certain tissue rather than fibrous tissue. The only exception may occur in the case of (cell-free) scaffold implantation next to some damaged tissue. The tissue restoration could be expanded to collagen area. The data related to cell attraction from a healthy tissue (including cartilage) is limited and often the effect of the tissue damage (e.g., in the course of scaffold implantation) still could be considered. In our study there was no such a damaged tissue and the cartilage was formed de novo in the area where there was no cartilage before.
- There is quantitative data for the in vivo study. Images based on one animal are not representative.
Histological and immunohistochemical studies conducted in the research could be considered as the most appropriate evidence of the processes but they still are qualitative. Deeper (quantitative) study is planning to carry out in the near future. As for a small number of animals, we are limited with bioethics issue and the total number of animals (12 – by 2 on each time point) was approved according to the study aim and methodology.
- Figure 3 has to be revised. It's unclear what it is showing.
The figure was modified as well its description.
- The language in the manuscript has to be improved significantly
The text was reviewed and corrected.

Round 2
Reviewer 3 Report
Thanks for replying to some of my previous concerns; however, some critical points still remains unanswered.
Author Response
Dear Reviewer, We have revised our response, as well as the manuscript. We hope we were able to reasonably answer all your comments. Yours faithfully, Research Team
Round 3
Reviewer 3 Report
Thanks for replying point to point to my questions.